# Facial Expression Recognition from Multi-Perspective Visual Inputs and Soft Voting

**DOI:** 10.3390/s22114206

**Published:** 2022-05-31

**Authors:** Antonio A. Aguileta, Ramón F. Brena, Erik Molino-Minero-Re, Carlos E. Galván-Tejada

**Affiliations:** 1Facultad de Matemáticas, Universidad Autónoma de Yucatán, Mérida 97110, Mexico; aaguilet@correo.uady.mx; 2School of Engineering and Sciences, Tecnologico de Monterrey, Monterrey 64849, Mexico; 3Departamento de Computación y Diseño, Instituto Tecnológico de Sonora, Ciudad Obregón 85000, Mexico; 4Instituto de Investigaciones en Matemáticas Aplicadas y en Sistemas—Unidad Yucatán, Universidad Nacional Autónoma de México, Sierra Papacal, Yucatán 97302, Mexico; erik.molino@iimas.unam.mx; 5Unidad Académica de Ingeniería Eléctrica y Comunicaciones, Universidad Autónoma de Zacatecas, Zacatecas 98000, Mexico; ericgalvan@uaz.edu.mx

**Keywords:** machine learning, information fusion, facial expressions

## Abstract

Automatic identification of human facial expressions has many potential applications in today’s connected world, from mental health monitoring to feedback for onscreen content or shop windows and sign-language prosodic identification. In this work we use visual information as input, namely, a dataset of face points delivered by a Kinect device. The most recent work on facial expression recognition uses Machine Learning techniques, to use a modular data-driven path of development instead of using human-invented ad hoc rules. In this paper, we present a Machine-Learning based method for automatic facial expression recognition that leverages information fusion architecture techniques from our previous work and soft voting. Our approach shows an average prediction performance clearly above the best state-of-the-art results for the dataset considered. These results provide further evidence of the usefulness of information fusion architectures rather than adopting the default ML approach of features aggregation.

## 1. Introduction

The use of the human face as a biometric means of identification—commonly called “face recognition” [1]—is currently widely used at the commercial scale, in devices ranging from cellphones to residential gateways [2], to the point that its use without people’s awareness has been called a threat to personal privacy [3]. Another potentially very helpful sub-area of face analysis is emotion recognition using facial expressions [4,5]. Of course, facial expressions are not direct indicators of subjective emotions for several reasons, starting with faked smiles or other expressions; current pre-trained facial expression recognition systems are unreliable when exposed to different individuals. The latter is why attributing emotions to specific individuals has been signaled as an unethical use of AI in the workplace [6]. Though many emotion recognition works have explored several cues beyond visual ones (such as speech [7], body gestures [8] and others), visual-related facial emotion recognition will remain one of the primary emotion recognition approaches for a long time. Facial expression recognition through visual analysis is certainly poised to make significant strides in the following years, mainly because of its great potential in real-world applications, even when used anonymously, from stores’ shop windows seeing customer reactions to engagement assessment in public events; the area’s financial value is expected to grow to over 40 billion dollars in the next five years.

This paper addresses the problem of automatic facial expression recognition and proposes a method based on information fusion and ML techniques. Our work builds on the previous work of Freitas et al. [9], which used a variant of visual expression recognition, namely, a set of facial points delivered by a Microsoft Kinect device [10]. We obtained even better results by applying the information fusion methods described below. The specific application of the dataset we used [9] was the recognition of facial expressions as a complement to hand and body gestures in sign language—specifically, the “Libra” Brazilian sign language. We will describe below in detail the specific settings and methods of this work to promote reproducibility.

Our proposed method consists of using subsets of the feature vectors mentioned above as independent feature vectors (which we call “perspectives”), from which several instances (one for each subset) of some classifier can learn to predict a facial expression (in terms of class probabilities). The predictions of such cases are processed by using soft voting [11] for the final decision. This approach has not been proposed previously for facial expression classification. As far as we know, the feature vector elements (coming from the same source) have not been treated as independent feature vectors (as if they came from different sources).

Like many other recent works on facial expression recognition [12,13,14], we leverage machine learning (ML) methods [15]. Instead of relying on human-defined rules, ML systems are entirely data-driven and can adjust their behavior mainly by training with a different dataset. Nevertheless, most ML works use dataset features as a flat vector (we call this approach “aggregation”), which could be sub-optimal for classification performance. In our previous works [16,17], we have explored the use of structured information combination architectures, such as separating the features (columns of the dataset) into groups and then applying hard or soft voting [11] and other methods for combining the predictions of instances of ML classifiers, one for each feature group. Though it could not be intuitively evident, the use of fusion, as mentioned earlier, gives in some cases substantially better results, in terms of accuracy and other quality indicators, than simple aggregation.

The contributions of this work are twofold: (1) a novel and efficient approach based on information fusion architectures and soft voting for the visual recognition of facial expressions, and (2) this approach improves the indicators of critical performance, such as accuracy and F1-score, compared to other state-of-the-art works, which studied the same dataset as us, as a result of exploiting information fusion architectures and soft voting with subsets of features.

This document is organized as follows: After this introduction, Section 2 establishes some definitions, and Section 3 reviews the main related works; then, Section 4 shows the proposed method; then, Section 5 presents the experimental methodology, and Section 6 discusses the results. Finally, Section 7, draws conclusions and suggests possible future work.

## 2. Background

This work lies in the intersection of two areas: one is the application area, which is facial expression recognition, and the other one is information fusion architectures for machine learning, which refers to the way input information is structured to get classification performance that is as high as possible. As far as the latter is concerned, we have previously done some work applying fusion architectures to domains such as activity recognition [17]. However, we were interested in testing our methods in a domain radically different from the activity recognition one, so facial expression recognition was a good candidate. Then, as we mentioned in the introduction, for the facial expression recognition task, we restricted our attention to facial expressions used intended to complement the gestures in sign languages, giving a prosodic component to the sentences [18,19]; this is why they have been called “grammatical facial expressions” (GFE). In recent years, GFE has gained importance in automated recognition tasks for sign languages, as they help eliminate confusion among signs [20]. GFE help with the construction of the meaning and cohesion of what is signed [9]; for example, they help to construct different types of phrases: questions, relative, affirmative, negative, and conditional.

GFE has been used in data-driven machine learning approaches for various sign languages, such as American (ASL) [21,22,23], German [24], Czech [25], and Turkish [26], among others. In the literature, it has been proposed that a classifier should learn to recognize syntactic-type GFE in “Libras” sign language (Brazilian sign language) using a vector of features composed of distance, angles and deep points, extracted from the points of the contour of the face (which were captured by a deep camera) [9]. In this paper, we are using the dataset proposed in Freitas’ work.

In addition, GFE has begun to be processed by taking advantage of data fusion techniques [27], such as, in the context of GFE recognition, combining the outputs of Hidden Markov Models (HMM) [28] (the probabilities of movements of facial features and movements head) and using them as input to a Support Vector Machine (SVM) [29], proposed by [30]. Kumar et al. [31], for their part, followed a similar approach to the previous one, where they used two HMMs as temporal classifiers to combine their decisions (facial gesture and hand gesture) through the Independent Bayesian Classification Combination (IBCC) method [32]. In addition, da Silva et al. [33] presented a model composed of a convolutional neural network [34] (to obtain the static features of two regions of the face image) and two long-short term memory networks [35] (to add the temporal features to the features of each face region), which ultimately merge their outputs for a final decision. Additionally, Neidle [36] described a majority voting strategy [11] that combines the SVM classifier trained with the eye and eyebrow region features and the angle of 100 inclination of the head. However, although these fusion techniques have shown promising results in GFE recognition, the use of information fusion techniques has been ad hoc and not systematic. We found no works that use such techniques in the particular case of Libras GFE recognition. In the context of GFE recognition, although the knowledge acquired in one sign language can be considered in others, it is necessary to study each of them separately, as they have their particularities [9]. The GFE facial expression set is specific for each signal language.

The GFEs we are considering in this paper aim to identify different types of sentences [37], which are the nine following ones used in the sign language of Libra [37,38,39]:**WH question**—phrases (such as who, when, why, how, and where) expressed by a slight elevation of the head, accompanied by lines drawn on the forehead.**Yes/No question**—interrogative sentences (in which there is a Yes or No answer) expressed with the head down and the eyebrows raised.**Doubt question**—sentences (showing distrust) expressed by compressing the lips, closing the eyes more, drawing lines on the forehead, and tilting the shoulders to one side or back.**Negation**—sentences (constructed with elements no, nothing, never) are drawn by lowering the corners of the mouth, accompanied by lowering of the eyebrows and lowering of the head, or a side-to-side movement of the head.**Affirmative**—phrases (that transmit ideas or affirmative actions) are expressed by moving the head up and down.**Conditional**—clauses (indicating that a condition must be met to do something) characterized by tilting the head and raising the eyebrows, followed by a set of markers that can express a negative or affirmative GFE.**Relative**—clauses (that add phrases either to explain something, add information, or insert another relative, interrogative sentence) are presented by raising eyebrows.**Topics**—serve to structure speech differently, such as moving an element (topic) of the sentence to the beginning. One way to express these sentences is by raising the eyebrows, moving the head down and to the side, and keeping the eyes wide open.**Focus**—Sentences that insert new information into speech, such as contrasting ideas, giving information about something, or highlighting something. These sentences are expressed in the same way as a topic sentence.

### 2.1. Data Fusion Architectures

The fusion of data from various sources (sensors) emerges from the observation that one data source can compensate for other data sources’ weaknesses, so with the combination of several sensors, it is possible to achieve better reliability, accuracy, flexibility, or combinations of the above; that is why the fusion of information from several sensors is currently used in many systems spanning many domains [40].

There are many ways to implement the general idea of data fusion. First of all, three different “levels” of fusion have been distinguished [41,42]:Fusion at the “data level” consists of gathering compatible data from sensors that could be different, but the incoming data are of the same type so they can be put together; this form of fusion is aimed at coverage, redundancy reliability, and increasing the amount of data.In the fusion at the “feature level”, the characteristics (“features”) extracted from different data sources, and usually of various types, are used to complement the other available ones, generally aiming at improving the accuracy or similar prediction quality metrics.At the “decision level” fusion, several independent predictions are obtained using some of the data or features, and then the partial decisions are combined by an algorithm like voting.

In practical systems, two or all of these fusion levels are often used, being combined in structures called “fusion architectures.” Aguileta [16] compared, in tasks such as activity recognition, the performance of several fusion architectures, including the following ones:Raw feature aggregation, which is a kind of baseline with almost no structure: It is simply concatenating the columns of several datasets with compatible rows (there could be some issues to sort out, such as if the clocks of sensors in a time series are not perfectly aligned, if there are missing data from one of the sensors, etc.). Raw feature aggregation is one of the simplest, “no structure” options.Voting with groups of features by sensor and homogeneous classifier. This architecture takes the features from each sensor and uses them to train a respective ML classifier (in this case, the classifiers are the same, such as random forest, for all sensors); then, we combine the classifier predictions using voting.Stacking with shuffled features: We shuffle the features randomly, and then we partition them into equal parts, which group the columns of the dataset, and then we train independent (usually similar) classifiers with each group. Then the predictions of each classifier become a feature in a new dataset, for which we train a classifier that we use to make the actual prediction.

The last two architectures are just examples from our previous work [16]. However, it should be clear that the number of possible architectures is staggering because they are combinations of structural elements, such how we group the features, which classifier are we going to use for each one (and whether or not it should be the same one), how to combine the classifiers’ decisions, and so on.

In this paper, we do not explore the problem of choosing the best architecture for a given dataset, which has been done elsewhere [16]. However, we do establish that the result, using a non-trivial architecture in the domain we are considering, is better than simple aggregation in terms of performance measures to a statistically significant extent.

### 2.2. Soft Voting

In the previous subsection, we have mentioned voting inside of fusion architectures, but we must further distinguish between two voting variants: hard and soft voting.

Hard voting, also called simply “voting” or “plurality voting”, is what we usually call “voting”: the choice receiving more votes is the one chosen. However, in “soft voting” there are weights for each vote that are taken into account. A weighted linear average is calculated and compared to a predefined threshold, giving the final result [43].

In the case of ML systems, the weights usually are taken from the certainty given as a percentage by the classifier about a given decision. Though roughly the certainty is supposed to correspond to a probability, most of the implemented methods in commonly used software packages are not strictly probabilities, so they should be used cautiously.

Section 4 will explain how we use soft voting to achieve better performance than with hard voting.

## 3. State of the Art

In works addressing GFE recognition related to Libra sign language (using a dataset for Brazilian sign language [37]), Bhuvan et al. [44] explored various machine learning algorithms (such as the multi-Layer perceptron (MLP) [15], the random forest classifier (RFC) [45], and AdaBoost [46], among others) to recognize nine GFEs. They performed experiments (with the 100 coordinates (x,y,z) corresponding to facial points stored in the aforementioned dataset) under the user-dependent model (when training and prediction of a classifier are performed with the same subjects) to choose the best algorithm for each GFE. The primary metrics on which they based these choices were the area under the curve (AUC) of the receiver operating characteristic (ROC) [47] and the F1 score [48].

Acevedo [49] applied morphological associative memories (MAMS) [50] to recognize nine GFE. They performed experiments with the 100 coordinates (x,y,z) corresponding to the facial points stored in the aforementioned dataset for both subjects (one and two). MAMS performance was measured with the % error and its complement (% recognition).

Gafar [51] proposed a framework to recognize nine GFE. It relies on two algorithms to reduce features and the fuzzy rough nearest neighbor (FRNN) [52,53] algorithm (which is based on the k-nearest neighbor [54] algorithm) for the classification task. These two algorithms (called FRFS-ACO [55,56], when used together) are the fuzzy rough feature selection (FRFS) [57,58] algorithm and the ant colony optimization (ACO) [59,60] algorithm. He performed experiments with the 100 coordinates (x,y,z) corresponding to the facial points stored in the previous dataset for subject one. The framework’s performance, which was compared with others (such as FRFS-ACO with MLP, FRFS-ACO with C4.5 [61], and FRFS-ACO with fuzzy nearest neighbor (FNN) [62]), was measured with the accuracy metric [15].

Uddin [20] presented an approach based on two methods (AdaBoost and RFC) to recognize nine GFE. The AdaBoost feature selection algorithm was used to reduce features and RFC for the classification task. He performed experiments with the 100 coordinates (x,y,z) corresponding to the facial points stored in the previous dataset for subject one and subject two. The approach performance was measured with the AUC-ROC metric.

Freitas et al. [9] used MLP to recognize nine GFEs. They performed experiments with the 100 coordinates (x,y,z) corresponding to the facial points stored in the previous dataset for both subjects (one and two). These experiments mainly involved creating a feature vector (composed of the distances, angles, and coordinates, extracted from said points), different sliding window [63] sizes to add the time feature to said feature vector, and various training and testing strategies. Based on the user-dependent model and the user-independent model (when training and predictions of a classifier are carried out with the different subjects), some examples of these strategies are (1) training and validation with subject one or two and testing with subject one or two, and (2) training and validation with subjects one and two and testing with the same two subjects. MLP was measured with the F1 score.

Cardoso et al. [64] classified six GFEs using MLP. They used eight points $\left(x_{i},y_{i}\right)$ of the face, which together with the distances between them, formed the characteristics of the GFE. For the experiment, they used the user-dependent model and the user-independent mode. The results of the experiments were presented as accuracy.

Our work differs from previous work, as we consider different subsets of the feature vector (extracted from the Libra sign dataset) as independent feature vectors to take advantage of fusion techniques (such as soft voting). As we have shown, such a strategy has not been explored in previous works. Additionally, in user-dependent experiments (see Section 5), we used the same sliding window size for all GFE studied here, unlike previous works.

## 4. Method and Materials

The approach we propose is illustrated in Figure 1. It takes advantage of the data fusion strategy in a context where a sequence of data over time maintains a meaning for a given period, such as GFE. This approach consists of four steps that we describe below:

In step 1, we extract from the raw data (for example, the *X*, *Y*, and *Z* coordinates that represent the human face in each unit of time, for a given period of time) three features (such as distances, angles, and *Z*s, which have been used with good results in this task [37]). Formally, let $FE=(p_{1},\ldots ,p_{n})$ be a set of *n* points that represent a facial expression with $p_{i}=\left(X_{i},Y_{i},Z_{i}\right)\in R^{3}$ for i=1,…,n. Then, taking the *X* and *Y* of some FE points, we define a set of pairs of points (from which we calculate the Euclidean distances) as $PP=\{pp_{1},\ldots ,pp_{l}\}$ for a given *l*, where $pp_{k}=\{\left(X_{(k*2)-1},Y_{(k*2)-1}\right),$$\left(X_{k*2},Y_{k*2})\right\}$ for k=1,…,l. Therefore, the Euclidean distance feature is defined as the set $DF=(d_{1},\ldots ,d_{l})$ where the Euclidian distance $d_{k}=ED\left(pp_{k}\right)$ (see Equation (Equation 1)).
(1)$$ED\left(pp_{k}\right)=\sqrt{|X_{(k*2)-1}-X_{i*2}{|}^{2}+{\left|Y_{(k*2)-1}-Y_{i*2}\right|}^{2}}$$

Additionally, taking the *X* and *Y* of the FE points, we define a set, whose elements are formed by three points, from which we calculate angles. This set is $PPP=\{ppp_{1},\ldots ,ppp_{m}\}$ for m<n, where $ppp_{j}=\{(X_{(j*3)-2},$$Y_{(j*3)-2}),\left(X_{(j*3)-1},Y_{(j*3)-1}\right),$$\left(X_{j*3},Y_{i*3})\right\}$ for j=1,…,m. Therefore, the angle feature is defined as the set $AF=(a_{1},\ldots ,a_{m})$, where $a_{j}=AN\left(ppp_{j}\right)$ (see Equation (Equation 2)).
(2)$$AN\left(ppp_{j}\right)=tan^{-1}\left[\frac{\frac{Y_{(j*3)-2}-Y_{(j*3)-1}}{X_{(j*3)-2}-X_{(j*3)-1}}-\frac{Y_{(j*3)-1}-Y_{i*3}}{X_{(j*3)-1}-X_{j*3}}}{1+\frac{Y_{(j*3)-1}-Y_{i*3}}{X_{(j*3)-1}-X_{j*3}}-\frac{Y_{(j*3)-2}-Y_{(j*3)-1}}{X_{(j*3)-2}-X_{(j*3)-1}}}\right]$$

Taking as reference the non-repeated PP points, we take their corresponding Zs located in the FE set and define the set $ZF=(Z_{1},\ldots ,Z_{ll})$, which corresponds to the third feature, where ll⩽l.

In step 2, by adding the temporal characteristic to the features we defined above, we create three sets of features or “perspectives” (as we call them). The temporal characteristic is added to these features by observing a series of consecutive facial expressions in time, which slide one expression at a time (“sliding window” procedure [37]), where a GFE is supposed to occur. Formally, let sw be the size of the window of the facial expressions (the number consecutive facial expressions included in a window) and sfe the number of facial expressions. Then, the first “perpective” is defined by
$$P1=\{VF1^{1},\ldots ,VF1^{sfe-sw}\}$$
with the set VF1 defined in Equation (Equation 3),
(3)$$\begin{array}{ccc} \;VF1^{t} & = & (DF^{t},AF^{t},ZF^{t},DF^{t+1},AF^{t+1},ZF^{t+1},\\ \; & \; & \ldots ,DF^{\frac{sw}{3}-1+t},AF^{\frac{sw}{3}-1+t},ZF^{\frac{sw}{3}-1+t}) \end{array}$$

The second “perspective” is defined by
$$P2=\{VF2^{1},\ldots ,VF2^{sfe-sw}\}$$
with the set VF2 defined in Equation (Equation 4),
(4)$$\begin{array}{ccc} \;VF2^{t} & = & (DF^{\frac{sw}{3}+t},AF^{\frac{sw}{3}+t},ZF^{\frac{sw}{3}+t},DF^{\frac{sw}{3}+t+1},\\ \;\; & \; & AF^{\frac{sw}{3}+t+1},ZF^{\frac{sw}{3}+t+1},\ldots ,DF^{\frac{2*sw}{3}-1+t},\\ \; & \; & AF^{\frac{2*sw}{3}-1+t},ZF^{\frac{2*sw}{3}-1+t}) \end{array}$$

The third “perspective” is defined by
$$P3=\{VF3^{1},\ldots ,VF3^{sfe-sw}\}$$
with the set VF3 defined in Equation (Equation 5),
(5)$$\begin{array}{ccc} \;VF3^{t} & = & (DF^{\frac{2*sw}{3}+t},AF^{\frac{2*sw}{3}+t},ZF^{\frac{2*sw}{3}+t},\\ \;\; & \; & DF^{\frac{2*sw}{3}+t+1},AF^{\frac{2*sw}{3}+t+1},ZF^{\frac{2*sw}{3}+t+1},\\ \; & \; & \ldots ,DF^{sw-1+t},AF^{sw-1+t},ZF^{sw-1+t}) \end{array}$$

In the $VF1^{t}$ set, $VF2^{t}$ set, and $VF3^{t}$ set, the set $DF^{t}=\left(d_{1}^{t},\ldots ,d_{l}^{t}\right)$ is the set DF calculated with the points $pp_{k}$ extracted from FE in the time t=1,…,sfe−sw. Additionally, the set $AF^{t}=\left(a_{1}^{t},\ldots ,a_{m}^{t}\right)$ is the set AF calculated with the points $ppp_{j}$ extracted from FE in the time t=1,…,sfe−sw. The set $ZF^{t}=\left(Z_{1}^{t},\ldots ,Z_{l}l^{t}\right)$ is the set ZF referenced by the points $pp_{k}$ extracted from FE in the time t=1,…,sfe−sw.

In step 3, we learn to predict a GFE from three classifiers (such as RFC), one for each “perspective” (P1, P2, and P3). Here, the P1 set, together with its corresponding labels $L1=(l_{1},\ldots ,l_{sfe-sw})$, are the input for a RFC instance, which predicts the probability of a GFE label. Similarly, the P2 and P3 sets, and their corresponding set of labels $L2=(l_{1},\ldots ,l_{sfe-sw})$ and $L3=(l_{1},\ldots ,l_{sfe-sw})$, respectively, are the inputs of independent RFC instances, which predict the probability of the tag of the same previous expression. Here, pij is the probability of the class i=1,2 (for a binary classification) predicted by the classifier j=1,2,3 (for the three classifiers).

Finally, in step 4, the final decision, $\hat{y}$, is taken by soft voting of the predictions of the classifiers of the previous step using Equation (Equation 6) [11]:(6)$$\hat{y}=arg\;\underset{i}{max}\sum_{j=1}^{3}w_{j}p_{ij},$$
where $w_{j}$ is the weight of the *j*th RFC instance. In this case the weights are taken uniformly because we use the same classification algorithm (RFC) instances.

## 5. Experimental Setup

Here, we conducted four types of experiments following a similar procedure as the one proposed by [37]: (1) The first experiment consisted of training and testing our model with data from subject one only; (2) In the second experiment, training and testing were done using data from subject two only; (3) For the third experiment, training and testing were done with data from both subjects. These three experiments were subject-dependent, and for evaluation of the trained models, k-fold cross-validation was used, with k = 10; (4) The fourth and last experiment followed a subject-independent strategy; here the entire dataset from subject one was used for training, and the total dataset from subject two was used for testing. The accuracy metric was used to compare results with other approaches described in the literature, in the experiments of the first type, and the F1-score and the AUC-ROC in all four experiments.

### 5.1. Datasets

The data with which our proposed method was tested corresponds to the Grammatical Facial Expressions set [37]. A complete description is available at the UCI machine learning repository [65]. This set consists of 18 videos of facial expressions from two subjects. Each subject performed five sentences in the Brazilian sign language that require a facial expression. The data were acquired through a Microsoft Kinect sensor that produced 100 spatial data points $\left(x_{i},y_{i},z_{i}\right)$, numbered from 1 to 100, from different face parts—the eye, mouth, nose, etc. Each frame has a binary class labeled by an expert, corresponding to positive (P) for the case where the expression is present, or negative (N) for no expression. Nine different facial expressions were mapped: affirmative, conditional, interrogative doubt, emphasis, negative, relative, topic, interrogative questions, and interrogative yes/no questions.

### 5.2. Feature Extraction and Perspective Construction

From the dataset proposed by [37], we took different coordinate points of the face for our experiments. In user-dependent experiments, the face landmarks we used were points from the left eyebrow (18, 22, and 24), right eyebrow (31 and 34), left eye (3 and 6), nose (39, 42, and 44), and mouth (49, 56, 59, 60, 62, 63, 64, and 66). In user-independent experiments, the face points we used were the left eyebrow (17, 22, and 24), right eyebrow (27, 31, 32, and 34), left eye (0, 2, and 6), right eye (8, 9, 10, 14, and 15), nose (89), and mouth (60, 61, 62, 63, 64, 66, and 67).

From these coordinate points, we extracted three types of features: distances (see Equation (Equation 1)), angles (see Equation (Equation 2)), and a temporal parameter to build a perspective of expression over time, which corresponds to features of a range of frames within a window. These were then concatenated with all the corresponding features from the following frames within a window (see Section 4). For user-dependent experiments, we used a 10-frame window in each GFE. For the subject-independent experiment, the number of windows was different for each expression: affirmative: 4; conditional: 10; doubts: 5; emphasis: 10; relative: 10; topic: 10; wh-questions: 2; negative: 6; and yes/no questions: 10. In all cases, windows overlap by one frame.

For the doubt expression case, we found that the distance and angle features presented small changes between cases where labels were positive or negative, mainly due to the type of expression which is characterized by a slight contraction of the eyes and mouth; thus, we included two extra features for the left eyebrow (90, 91, 92, 93, 94, 21, 22, 23, 24, 25), mouth (48, 49, 50, 51, 52, 53, 54, 55, 56, 57, 58, 59), and left eye (0, 1, 2, 3, 4, 5, 6, 7), which are the enclosed area [66] and the principal axes ratio or eccentricity [67], which allowed us to better identify the characteristic patterns of this expression. Additionally, for the doubt expression case, we did not concatenate the features that belong to each frame in a window. Instead, we statistically described the features in that window. Therefore, for each feature (distances, angles, areas, and principal axis ratio) extracted from a frame and observed in a window, we calculated the following statistical descriptors: mean, standard deviation, maximum value, minimum value, median, 25th percentile, and 75th percentile, all of which we concatenated. In all cases, the features were normalized.

Since the number of observations for each expression is different, classes were balanced to prevent classification issues and bias in the direction of the larger class [68]. For this, the larger class was subsampled by random elimination of observations to equal that of the minor class.

Finally, we divided the features described above into three parts. Each of these parts was introduced in one of the RFC instances to use its results in the weighted voting of the final decision.

## 6. Results and Discussion

This section presents and discusses the results achieved by our proposed method when trained and tested with different subjects, in terms of three metrics: the F1-score, the accuracy, and the ROC-AUC. We also compare these results with the results obtained by the methods presented in the state-of-the-art section, which used the dataset studied here.

### 6.1. Results of Training and Testing with Subject One

In Table 1, we can see that our model reached an average F1-score of 0.99 for the GFE prediction of the Brazilian sign language (Libras) signs performed by subject one when our model was trained with these same types of expression made by the same subject (user-dependent approach). Additionally, we note that our method reached an average F1-score of 1 for 6 of the 9 GFEs from subject one. Additionally, in Table 1, we can see that our proposal outperformed the approaches proposed in the literature and studied here, from the perspective of the F1-score metric, for all grammatical facial expressions performed by subject one.

In Table 2, we can see that our model’s average accuracy exceeded 99 percent for GFE prediction. Besides, from the perspective of the accuracy metric, we note that our proposal bettered the approaches from the state-of-art studied here, in all the GFEs cases.

Table 3 also shows very good performance from our model, with an average ROC-AUC of 0.9997. In Table 3, we observe that our proposal bettered the approach of Bhuvan et al., from the perspective of the ROC-AUC metric, for all GFEs reviewed here and performed by subject one.

Table 1, Table 2 and Table 3 indicate that our method has great potential for the user-dependent approach, at least for subject one of the GFE dataset studied here. Additionally, These tables suggest that our process learns better GFEs under the user-dependent approach to subject one than the state-of-art models reviewed here.

### 6.2. Results of Training and Testing with Subject Two

In Table 4, we can see the good performance of our model, with an average F1-score of 0.99, for the prediction of GFEs under the user-dependent approach for subject two. Our proposal bettered the state-of-art techniques studied here, from the perspective of the F1-score metric, for all GFEs.

Table 5 also shows the good performance from our model for GFE prediction, with an average ROC-AUC of 0.9997, under the user-dependent approach for subject two. Additionally, we note that our proposal outperformed the method of Bhuvan et al., from the perspective of the ROC-AUC metric, for all GFEs.

Table 4 and Table 5 show that our method has great potential for the user-dependent approach for subject two of the GFE dataset studied here. Furthermore, these tables suggest that our method recognizes GFEs from subject two better than the state-of-the-art models reviewed here.

### 6.3. Results of Training and Testing with Subject One and Subject Two

In Table 6, we can see that our model reached an average F1-score of 0.99 for the GFE prediction of the Brazilian sign language (Libras) expressions made by subject one and subject two when our model trained with these same types of expression performed with the same two subjects (user-dependent approach).

From the perspective of the F1-score metric, Table 6 shows that our proposal bettered the approaches proposed in the literature (considered here) for all grammatical facial expressions performed by subject one and subject two.

In Table 7, we can see that our model reached an average ROC-AUC of 0.9996 for predicting GFEs under the user-dependent approach for subject one and subject two. Further, we note that our proposal is better or equal to the state-of-art methods analyzed here, from the perspective of the ROC-AUC metric, when GFEs are analyzed separately. On the other hand, when GFEs were analyzed together, our method bettered cutting-edge approaches with an average ROC-AUC of 0.9996.

The results in Table 6 and Table 7 suggest that our method performs well in the user-dependent approach for subject one and two of the GFE dataset studied here. Furthermore, these results indicate that our method better identifies GFE under the user-dependent approach for the two subjects jointly studied here than the approaches proposed in the literature.

### 6.4. Results of Training with Subject One and Testting with Subject Two

In Table 8, we observe that under the user-independent approach, we achieved an average F1-score of 0.8420. These results suggest that our model can generalize well in this user-independent case.

Furthermore, in Table 8, we can observe that our model achieved, on average, better results than the state-of-the-art. We emphasize that, in six expressions (doubt question, affirmative, conditional, relative, topic, and focus), our model beat the results of state-of-the-art methods. We also noted that our approach performed slightly less than the state-of-the-art for the Y/N question expression. Additionally, we observed that our approach was not as good as the state-of-the-art in two expressions. These results suggest that, for the user-independent approach, our method generalizes better in six out of nine GFE than the methods proposed in the literature for the dataset studied here. However, overall, the results suggest that our method is better than the methods proposed in the literature.

Finally, based on the above results, our method can predict nine GFEs of the Brazilian Sign Language (Libras) very well under the user-dependent approach and six of these nine GFEs from good to reasonably well under the user-independent approach. The results above support our claim that our method predicts these nine GFEs more accurately than the state-of-the-art approaches studied here, from the perspective of three metrics (for the user-dependent case): F1-score, accuracy, and ROC-AUC. Furthermore, our method achieved superior results for these six GFEs, in terms of the F1-score, for the case of the user-independent approach, compared to the results of the methods proposed in the literature.

## 7. Conclusions

This paper proposed an improved method for recognizing facial expressions, among a collection of nine GFEs of the Brazilian sign language (Libras), from the visual face information composed of points delivered by a Kinect sensor. Our method is based on an information fusion approach which groups in a multi-view fashion the features (extracted from diverse points of the face) and then applies a decision-making architecture based on soft-voting to the outputs of various RFC instances. Thus, each view (one subset of the feature set) is used to train a classifier instance, and the prediction outputs of several instances are voted for the final decision of the GFE.

The results we presented in this paper show that our method is efficient and has better performance (considering three metrics: F1-score, accuracy, and ROC-AUC) than other state-of-the-art methods for the dataset considered.

Based on the results of the user-independent experiments and the user-dependent experiments’ results, we can make the claim of superior performance and hence an advance in recognizing facial expressions, at least for the dataset we considered (using Libras sign language), by using the multi-view architecture that we have also used in other domains, in combination with soft voting. We view this as an original contribution.

Our future work will address a more general problem of emotion recognition from the recognition of facial expressions, which of course, would have a greater commercial and social impact than the case of sign language, and some privacy implications that are better to consider from the initial design of the technology rather than making them an afterthought.

## Figures and Tables

**Figure 1 sensors-22-04206-f001:**
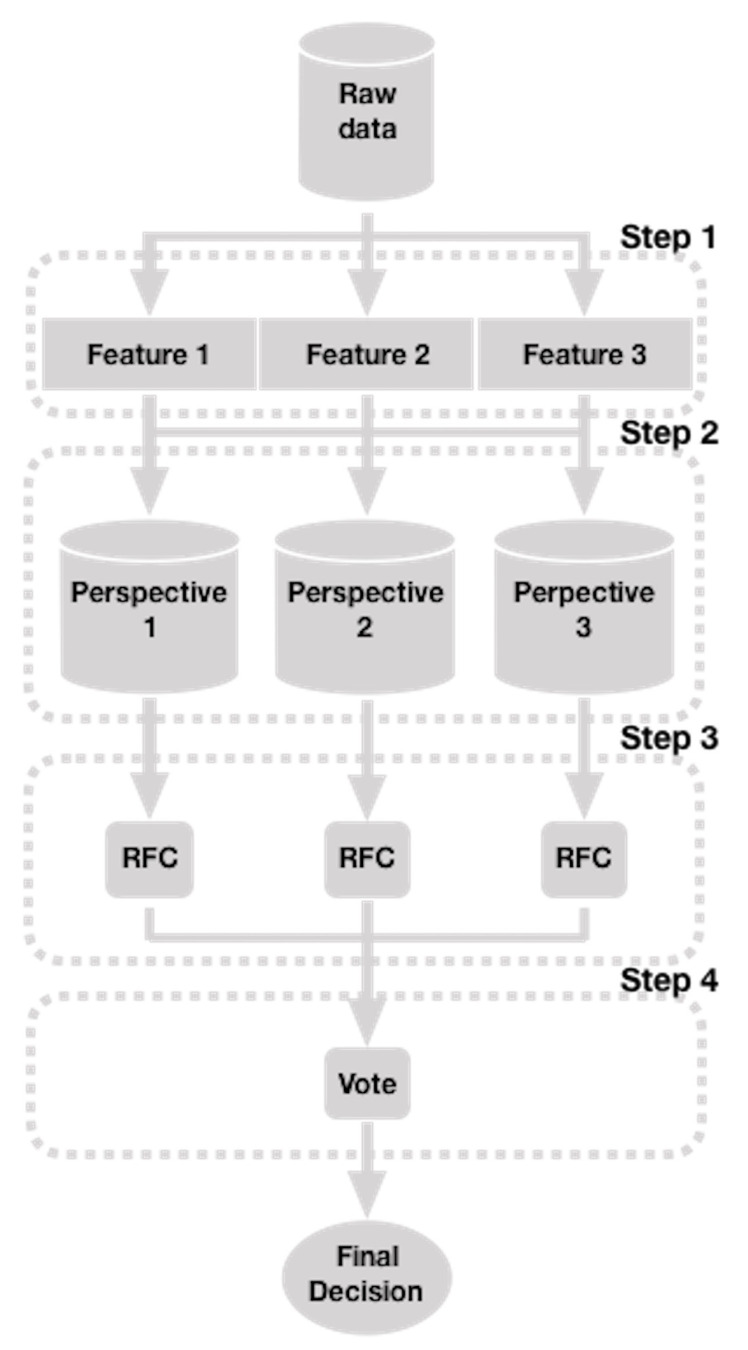
Overview of the method that predicts GFEs.

**Table 1 sensors-22-04206-t001:** F1 scores achieved by our proposed model and other state-of-art approaches using the dataset that stored grammatical facial expressions of the Brazilian sign language (Libras) made by subject one. Best results are in bold.

GFE	Freitas	Bhuvan	Our Proposal
Wh question	0.8942	0.945338	**0.98**
Y/N question	0.9412	0.940299	**0.99**
Doubt question	0.9607	0.898678	**1.00**
Negative	0.9582	0.910506	**1.00**
Affirmative	0.8773	0.890052	**0.98**
Conditional	0.9534	0.94964	**1.00**
Relative	0.9680	0.954064	**1.00**
Topic	0.9544	0.902439	**1.00**
Focus	0.9836	0.975	**1.00**
Average	0.9434	0.9296	**0.99**

**Table 2 sensors-22-04206-t002:** Accuracy achieved by our proposed model and other state-of-art approaches using a dataset storing grammatical facial expressions of the Brazilian sign language (Libras) made by subject one. Best results are in bold.

GFE	Gafar (FRFS-ACO and MLP)	Gafar (FRFS-ACO and FRNN)	Ours
Wh question	0.9237	0.9447	**0.9850**
Y/N question	0.9467	0.9438	**0.9875**
Doubt question	0.9329	0.9077	**0.9990**
Negative	0.9119	0.919	**0.9961**
Affirmative	0.8635	0.8983	**0.9777**
Conditional	0.9622	0.9701	**0.9991**
Relative	0.9665	0.9656	**0.9976**
Topic	0.9649	0.9532	**0.9986**
Focus	0.9593	0.933	**1.0000**
Average	0.936	0.9373	**0.9934**

**Table 3 sensors-22-04206-t003:** ROC-AUC scores achieved by our proposed model and other state-of-art approaches using a dataset that stores grammatical facial expressions of the Brazilian sign language (Libras) made by subject one. Best results are in bold.

GFE	Bhuvan et al.	Our Proposal
Wh question	0.9768	**0.9993**
Y/N question	0.9925	**0.9997**
Doubt question	0.9713	**1.0000**
Negative	0.9695	**1.0000**
Affirmative	0.9763	**0.9988**
Conditional	0.9915	**1.0000**
Relative	0.9946	**0.9999**
Topic	0.9863	**1.0000**
Focus	0.9948	**1.0000**
Average	0.9837	**0.9997**

**Table 4 sensors-22-04206-t004:** F1 scores achieved by our proposed model and other state-of-art approaches using a dataset storing grammatical facial expressions of the Brazilian sign language (Libras) made by subject two. Best results are in bold.

GFE	Freitas	Bhuvan	Ours
Wh question	0.8988	0.938776	**0.99**
Y/N question	0.9129	0.90566	**0.99**
Doubt question	0.9700	0.911765	**1.00**
Negative	0.7269	0.905556	**0.99**
Affirmative	0.8641	0.854772	**0.99**
Conditional	0.8814	0.867384	**0.98**
Relative	0.9759	0.935252	**0.99**
Topic	0.9322	0.853448	**0.99**
Focus	0.9213	0.934959	**1.00**
Average	0.8982	0.9008	**0.99**

**Table 5 sensors-22-04206-t005:** ROC-AUC scores achieved by our proposed model and other state-of-art approaches using a dataset storing grammatical facial expressions of the Brazilian sign language (Libras) made by subject two. Best results are in bold.

GFE	Bhuvan et al.	Our Proposal
Wh question	0.9872	**0.9999**
Y/N question	0.9754	**0.9998**
Doubt question	0.9697	**0.9999**
Negative	0.9749	**0.9993**
Affirmative	0.9485	**0.9996**
Conditional	0.9691	**0.9988**
Relative	0.9856	**0.9999**
Topic	0.9732	**0.9999**
Focus	0.9811	**1.0000**
Average	0.9739	**0.9997**

**Table 6 sensors-22-04206-t006:** F1 scores achieved by our proposed model and other state-of-art approaches using a dataset storing grammatical facial expressions of the Brazilian sign language (Libras) made by subject one and subject two. Best results are in bold.

GFE	Freitas	Bhuvan	Ours
Wh question	0.8599	0.925125	**0.99**
Y/N question	0.8860	0.922591	**0.99**
Doubt question	0.9452	0.928896	**1.00**
Negative	0.7830	0.909091	**1.00**
Affirmative	0.8209	0.898734	**0.98**
Conditional	0.8776	0.927176	**0.99**
Relative	0.8973	0.946087	**0.99**
Topic	0.9164	0.874109	**0.99**
Focus	0.9321	0.932462	**0.99**
Average	0.8798	0.9183	**0.99**

**Table 7 sensors-22-04206-t007:** ROC-AUC scores achieved by our proposed model and other state-of-art approaches using a dataset storing grammatical facial expressions of the Brazilian sign language (Libras) made by subject one and subject two. Best results are in bold.

GFE	Uddin	Bhuvan	Acevedo	Ours
Wh question	0.9853	0.9785	**1.0000**	0.9995
Y/N question	**1.0000**	0.9818	0.9594	0.9985
Doubt question	0.9833	0.9839	0.9500	**1.0000**
Negative	**1.0000**	0.9759	**1.0000**	0.9999
Affirmative	**1.0000**	0.9629	**1.0000**	0.9989
Conditional	0.9866	0.9835	0.9915	**0.9999**
Relative	0.9918	0.9935	**1.0000**	0.9999
Topic	0.9770	0.9728	**1.0000**	0.9999
Focus	0.9867	0.9874	**1.0000**	0.9999
Average	0.9901	0.9800	0.9890	**0.9996**

**Table 8 sensors-22-04206-t008:** F1 scores achieved by our proposed model and other state-of-art approaches that train a RFC with the Libras GFEs made by the subject one and test with the Libras GFEs of the subject two. Best results are in bold.

GFE	Freitas	Our Proposal
Wh question	**0.8743**	0.8409
Y/N question	**0.8365**	0.8346
Doubt question	0.9052	**0.9127**
Negative	**0.6760**	0.6667
Affirmative	0.7478	**0.7891**
Conditional	0.7704	**0.8014**
Relative	0.8653	**0.8694**
Topic	0.8953	**0.9168**
Focus	0.9022	**0.9463**
Average	0.8303	**0.8420**

## Data Availability

We made use of the publicly available dataset in https://archive.ics.uci.edu/ml/datasets/Grammatical+Facial+Expressions (accessed on 27 January 2022).

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
