# Peer review of "Facial Expression Recognition from Multi-Perspective Visual Inputs and Soft Voting"

_sensors, 2022, doi:10.3390/s22114206_

Round 1
Reviewer 1 Report
The study is an interesting one. "our approach shows an average 9
prediction performance clearly above the best state-of-the-art results for the dataset considered"- the authors should justify their point in a table with more studies if possible. They may incorporate precision and recall as well if possible.
Author Response
We thank you for your insightful review.
We added one more study that addresses the same data set that we do in the state of the art (see here the reference [1]). We did not compare ourselves with that study because they only considered six grammatical facial expressions (taken two by two and the neutral expression, for a multiclass classification), unlike us who analyzed nine.
In the future, we will study our approach with other data sets. Although precision and recall might provide more insight into our approach, we relied on the metrics that the other studies obtained for a benchmark.
[1] (or [64] in the paper) Cardoso, M.E.d.A.; Freitas, F.d.A.; Barbosa, F.V.; Lima, C.A.d.M.; Peres, S.M.; Hung, P.C. Automatic segmentation of grammatical facial expressions in sign language: towards an inclusive communication experience. Proceedings of the 53rd Hawaii International Conference on System Sci, 2020, pp. 1499–1508.
Reviewer 2 Report
A lot of work has been done recently in this area. The authors are kindly advised to refine the reference list with more recent publications published within the last five years. Currently, the list contains only 16 publications out of 60 published within the last five years. Some references could be therefore replaced with the most recent work.
Everything else is fine in my opinion.
Nice work!
Author Response
We thank you for your insightful review.
We added more references that are within the range of the last five years.
[1] (or [64] in the paper) Cardoso, M.E.d.A.; Freitas, F.d.A.; Barbosa, F.V.; Lima, C.A.d.M.; Peres, S.M.; Hung, P.C. Automatic segmentation of grammatical facial expressions in sign language: towards an inclusive communication experience. Proceedings of the 53rd Hawaii International Conference on System Sci, 2020, pp. 1499–1508.
[2] (or [31] in the paper) Kumar, P.; Roy, P.P.; Dogra, D.P. Independent Bayesian classifier combination based sign language recognition using facial expression. Information Sciences 2018, 428, 30–48. doi:https://doi.org/10.1016/j.ins.2017.10.046.
[3] (or [33] in the paper) da Silva, E.P.; Costa, P.D.P.; Kumada, K.M.O.; De Martino, J.M.; Florentino, G.A. Recognition of Affective and Grammatical Facial Expressions: A Study for Brazilian Sign Language. Computer Vision – ECCV 2020 Workshops; Bartoli, A.; Fusiello, A., Eds.; Springer International Publishing: Cham, 2020; pp. 218–236.
[4] (or [34] in the paper) Aggarwal, C.C.; et al. Neural networks and deep learning. Springer 2018, 10, 978–3.
[5] (or [35] in the paper) Yu, Y.; Si, X.; Hu, C.; Zhang, J. A review of recurrent neural networks: LSTM cells and network architectures. Neural computation 2019, 31, 1235–1270.
Reviewer 3 Report
The paper presents an ML ensemble model to identify different sentences using a Grammar Facial Expression approach. The Ensemble model precisely is a voting classifier that trains various random forest models and predicts on the basis of aggregating the findings of each random forest model.
This approach has been widely used in ML approaches for several years.
Concerning the article, I found major concerns:
- Structure: the paper's structure makes the reading hard. My suggestion is to clarify the problem in the introduction (describing the problem authors want to face and introducing the technologies). Then describe the SOA and only after that explain how GFE works.
- Some English grammatical errors are present in the paper. For instance:
"In practical systems, two or all of this fusion levels are often used, being combined in structures called “fusion architectures”. [15] compare in tasks like activity recognition the performance of fusion architectures like the following (among others):" Also, the sentence constructions should be revised. - I appreciate the methodology used for training and testing the ML model. The cross-database approach clarifies the results and highlights the model's accuracy.
Author Response
We thank you for your insightful review.
About point #1. We agree with the reviewer and as suggested we improved the Introduction by reorganizing the text and clarifying what is the problem we are addressing, (this can be found on lines 32 to 40 and 41 to 48).
Also, we changed the name of section 3 to SOA in order to clarify its scope.
We thoroughly considered the suggestion to change the order of sections 2 and 3, but we concluded that in order to provide a clear context for the SOA section, it is better to present first section 2 as background. We consider that with these changes the clarity of the paper has been improved.
About point #2, the article’s writing was thoroughly reviewed to correct English mistakes like the one that you highlighted. Over one hundred corrections were performed to polish the English writing.
Thanks for the positive feedback on point #3.
Reviewer 4 Report
a. Interesting and appropriate work for the special issue.b. There is a good background check and similar jobs.
c. The research design is appropriate and an adequate development of the project is presented.
d.The results presented are interesting and there is a correct discussion of them.
e. It is recommended to make the contribution clear in the title.
f. Justify the method of experimentation.
g. Update some references and add more recent citations.
Author Response
We thank you for your insightful review.
Thank you for the positive feedback in points (a) to (d).
We adjusted the title (to “Facial expression recognition from multi-perspective visual inputs and soft voting”) to comply with your recommendation (e), because the contribution of this work is a new improved method for facial expression recognition based on multi-view grouping of features as an information fusion structure that delivers better classification performance than SOA methods. Of course, it is difficult to convey all of this in a short title, we hope the adjusted proposed title fits the bill.
Regarding your point (f), our experimentation purpose was twofold. First, we wanted to know the performance of our proposal through the common metrics (Accuracy, F1-score, ROC-AUC). Second, we were interested in comparing our approach with other works in the same terms (user-dependent model and the user-independent model). Thus, we replicate the same experiments that the others proposed and also used their metrics because otherwise, direct comparison is not possible.
For your suggestion (g) we added more references that are within the range of the last five years.
[1] (or [64] in the paper) Cardoso, M.E.d.A.; Freitas, F.d.A.; Barbosa, F.V.; Lima, C.A.d.M.; Peres, S.M.; Hung, P.C. Automatic segmentation of grammatical facial expressions in sign language: towards an inclusive communication experience. Proceedings of the 53rd Hawaii International Conference on System Sci, 2020, pp. 1499–1508.
[2] (or [31] in the paper) Kumar, P.; Roy, P.P.; Dogra, D.P. Independent Bayesian classifier combination based sign language recognition using facial expression. Information Sciences 2018, 428, 30–48. doi:https://doi.org/10.1016/j.ins.2017.10.046.
[3] (or [33] in the paper) da Silva, E.P.; Costa, P.D.P.; Kumada, K.M.O.; De Martino, J.M.; Florentino, G.A. Recognition of Affective and Grammatical Facial Expressions: A Study for Brazilian Sign Language. Computer Vision – ECCV 2020 Workshops; Bartoli, A.; Fusiello, A., Eds.; Springer International Publishing: Cham, 2020; pp. 218–236.
[4] (or [34] in the paper) Aggarwal, C.C.; et al. Neural networks and deep learning. Springer 2018, 10, 978–3.
Round 2
Reviewer 3 Report
The authors clarified all the points highlighted in my first revision.